# Crystallization and Structural Properties of Oleogel-Based Margarine

**DOI:** 10.3390/molecules27248952

**Published:** 2022-12-15

**Authors:** Xiuhang Chai, Yujin Zhang, Yifei Shi, Yuanfa Liu

**Affiliations:** State Key Laboratory of Food Science and Technology, Collaborative Innovation Center of Food Safety and Quality Control in Jiangsu Province, School of Food Science and Technology, Jiangnan University, 1800 Lihu Road, Wuxi 214122, China

**Keywords:** natural waxes, oleogel, crystalline network, margarine, mouthfeel

## Abstract

Interest in oleogel as a promising alternative to traditional hydrogenated vegetable oil has increasingly grown in recent years due to its low content of saturated fatty acids and zero trans fatty acids. This study aimed to develop wax-based margarine to replace traditional commercial margarine. The wax-based margarine was prepared and compared with commercial margarine in texture, rheology, and microscopic morphology. The possibility of preparing margarine at room temperature (non-quenched) was also explored. The results showed that the hardness of oleogel-based margarine increased as the BW concentration increased. Denser droplets and crystal network structure were observed with the increase in BW content. XRD patterns of oleogel-based margarine with different content BW were quite similar and structurally to the β′ form. However, the melting temperature of oleogel-based margarine was over 40 °C at each concentration, which represented a poor mouth-melting characteristic. In addition, the unique, improved physical properties of oleogel-based margarine were obtained with binary mixtures of China lacquer wax (ZLW) and Beeswax (BW), due to the interaction of the ZLW and BW crystal network. The rapid cooling process improved the spreadability of oleogel-based margarine. The margarine prepared by 5% BW50:ZLW50 had similar properties to commercial margarine in texture and melting characteristics (37 °C), which had the potential to replace commercial margarine.

## 1. Introduction

With the improvement of living standards and food processing technology, margarine accounts for an increasing proportion of daily dietary consumption. Compared with natural butter, margarine has a relatively low cost and larger output, which can better meet the needs of marketization. In general, margarine is a water-in-oil (W/O) emulsion, consisting of 80~85% solid fat, 14–17% water, and a small amount of emulsifier [1]. The texture and stability of margarine during the storage period are affected by solid fat, which crystallized to trap the liquid oils and water droplets inside interspaces of the crystal network [2,3]. Nowadays, hydrogenated vegetable oil and palm oil are used in margarine as the most important solid fats, containing a lot of saturated fatty acids (SFAs) or even trans fatty acids (TFAs) in the diet [4]. However, many studies have proved that excessive intake of SFAs and TFAs increased the risk of cardiovascular and coronary heart disease, affecting human health [5]. Therefore, it is urgent to actively find a feasible way to replace the hard fat in traditional margarine.

Oleogel is a kind of supramolecular organic gel with liquid vegetable oil as the continuous phase, which can completely or partially replace traditional hard fats [6]. It does not contain TFAs, has a low proportion of SFAs, and is rich in monounsaturated and polyunsaturated fatty acids [7]. Wax as one of the important structuring agents had been used for preparing oleogels due to its high technological structuring capacity [8]. During the last decade, more research focused on the physicochemical properties of oleogels. Research showed that wax-based oleogels had unique advantages, such as high oil-binding capacity and low cost [9,10]. In addition, natural waxes may also lower serum cholesterol levels in the human body [11]. Recently, wax-based oleogels showed the potential to be used as margarine and shortening. Yilmaz and Ogutcu pointed out that the texture and sensory properties of sunflower wax (SFW) and beeswax (BW) oleogel-based shortening cookies resembled bakery shortening cookies [12]. Similar results were obtained from SFW, rice bran wax (RBW), and candelilla wax (CW)-based shortening cookies with desirable spreadability and soft eating characteristics [13,14]. In addition, Hwang et al. pointed out that margarine structured with SFW, RBW, and CW had better nutritional properties and appearance, but the taste and firmness need to be improved [8].

Despite the numerous studies, the biggest problem of wax oleogel-based margarine always showed its undesirable waxy mouthcoating, due to the higher melting points of wax. For example, margarine prepared using SFW oleogel with 2% or higher concentrations had higher melting points than that of commercial margarine [8]. Luckily, the research found that a wax mixture with eutectic melting properties could lower the melting point of waxes [10,15]. Hwang and Winkler-Mose pointed out that 3% CW-BW-based margarine had a similar firmness with commercial spread [16]. In addition, fewer blends of waxes could achieve higher gel strength, which is conducive to the sensory and melting properties of wax oleogel-based products. In our previous studies, we found China lacquer wax (ZLW) had a good mouthfeel, and a mixture of ZLW and BW had higher gel strength. The ZLW and BW oleogel showed preferable mechanical properties and mouthfeel [17]. However, the interaction of wax with the emulsifiers or water led to the different properties of wax-based oleogels appearing when they were used as margarine. For instance, RBW oleogels with soybean oil had a stronger network, whereas margarine structured with these oleogels was very soft [8]. Therefore, more information needs to be explored for their practical application.

This study aims to explore the effect of BW concentration on the physicochemical properties of margarine. Furthermore, the texture and microstructure of binary wax oleogel-based margarine with BW and ZLW were investigated, to explore the relationship between microstructure and macroscopic physical properties, providing information for the application of wax-based oleogel in food systems.

## 2. Results and Discussion

### 2.1. Hardness of BW Oleogel-Based Margarine

Hardness is an important index to evaluate the quality of margarine, which determines its culinary performance and spreadability. Our previous study reported that adding 5 wt% concentration BW into sunflower oil could form stable oleogels with needle-like crystals and order crystal networks, leading to a good high oil-binding capacity and rheological property [17]. Therefore, BW oleogels could be used to make healthy margarine with more unsaturated fatty acids.

Figure 1A shows the hardness of BW oleogel-based margarine and commercial margarine. The stable margarine was formed using BW oleogels with 4% concentration because of the higher gel-forming ability [18,19], and the hardness was 62 g. With the increase in BW concentration, the hardness increased significantly. When BW concentration increased to 10%, the value of hardness reached 200 g, which is almost four times bigger than that of 4% BW oleogel-based margarine. This is due to the formation of more crystals and a denser crystal network at higher wax content, resulting in the increased rigidity of BW oleogel-based margarine. In addition, the hardness of the non-quenched sample was higher than that of the quenched sample at the same concentration. Different processing conditions result in different crystallization behavior and crystal structure in BW oleogel-based margarine. The fast crystallization rate was instrumental in forming more fine crystals and more orderly crystal arrangement, leading to the difference in the hardness of margarine. In addition, the hardness of commercial margarine was different from that of wax oleogel-based margarine, with a value between 4% and 6% BW oleogel-based samples. The difference might be due to the difference between TAG crystallization and wax crystallization [20].

### 2.2. Rheological Properties of BW Oleogel-Based Margarine

To further understand the macroscopic properties of margarine, the rheological properties of BW oleogel-based margarine and commercial margarine were evaluated. Linear viscoelastic region (LVR) refers to the range of stress/strain within which the elastic/viscosity modulus is constant. As shown in Figure 1C–E, commercial margarine has the longest LVR, exhibiting appropriate brittleness and ductility [19]. The LVR of all the oleogel-based margarine was different with the difference of BW concentration due to the variance of internal structure in margarine. The results are in agreement with the hardness of oleogel-based margarine, as already mentioned in Figure 1. Small deformation occurred in the LVR, and subsequently, permanent deformation occurred with the decrease in elastic/viscosity modulus, leading to the breakage of margarine structure. 

Meanwhile, with the increase in the BW content, the LVR of oleogel-based margarine decreased gradually, which demonstrated that the brittleness of oleogel-based margarine samples increased accompanied by the increase in BW content, and the margarine prepared using a rapid cooling process was more brittle and exhibited poorer ductility. In addition, the LVR of margarine prepared under the rapid cooling process was significantly longer than that of margarine prepared without the rapid cooling process, which indicated that the quenching kneading process was conducive to the ductility and smearability of the product. 

Moreover, the value of G’ and G” obtained from the frequency scanning test reveal the elastic and viscous properties of the samples, when G’ > G”, the sample behaves as colloids (solid state). When G’ < G”, the sample behaves as a fluid [21]. Figure 1F–H shows the G’/G” changes of samples with the frequency sweep. G’ and G” are independent of frequency, and all the samples behave as solid (G’ > G”) with stronger interaction between crystal particles in the margarine. Commercial margarine was less frequency dependent than that of wax oleogel-based margarine, because of the formation of a stronger fat crystal network in commercial margarine [22]. With the increase in BW content, oleogel-based margarine was more frequency dependent. Compared with that of oleogel-based margarine prepared with a rapid cooling process, a higher G’ value of oleogel-based margarine prepared with no rapid cooling process was observed, which implied stronger solid properties. Thus, the oleogel-based margarine samples prepared with a rapid cooling process were more similar to the commercial margarine, while the oleogel-based margarine prepared with no rapid cooling process showed higher hardness and brittleness which indicated poor spreadable properties.

### 2.3. Solid Fat Content (SFC)

SFC is usually used to evaluate whether the sample has good mouth-melting properties for traditional special fats. Margarine with steep curves at the range of 10 °C~21.1 °C tended to exhibit better oxidation stability, while lower SFC at 33 °C was related to good mouth-melting properties. 

Figure 1B shows the SFC curves of oleogel-based margarine and commercial margarine at different temperatures. SFC decreased with the increase in temperature from 0 °C to 55 °C due to the melting of crystals in the margarine. Compared with oleogel-based margarine, higher SFC was observed at the lower temperature (0~25 °C) for commercial margarine, and a faster decline of SFC with an increase in temperature. This was because of the difference in crystal composition between the triglyceride crystals in commercial margarine and BW crystals in oleogel-based margarine. Since the oleogel-based margarine was structured by a small amount of solid BW crystals, its structural characteristics determined that oleogel cannot provide a large amount of solid fat. Moreover, the SFC of BW oleogel-based margarine increased gradually with the increase in BW composition, which indicated that BW crystals were the main source of SFC for oleogel-based margarine. For example, SFC values of oleogel-based margarine at 33 °C decreased from 7.5% to 0% with the decrease in BW content. BW oleogel-based margarine with 4% content had the lowest SFC with a value of 0%, indicating that no BW crystals existed in this system. SFC of oleogel-based margarine with 8% and 10% BW or BW’ was higher than that of commercial margarine, which would result in the waxy taste of oleogel-based margarine [23].

### 2.4. Melting Properties of BW Oleogel-Based Margarines

DSC was used to further determine the melting characteristics of different kinds of margarine, and the melting curves of BW oleogel-based margarine and commercial margarine are shown in Figure 2A. The oleogel-based margarine samples showed a wide melting range from 20 to 55 °C, because of different lipid components with different melting points in oleogel. Similar results were also reported in previous studies [13,24]. No significant differences were observed in the melting range (20 °C~55 °C) for oleogel-based margarine samples with different BW concentrations, indicating that BW determines the melting characteristics of oleogel-based samples, not the moisture and other solid components.

However, a different profile was observed in the commercial margarine with a value of 33.45 °C, while the peak temperature of all BW oleogel-based margarine was over 40 °C. As shown in Appendix A, the peak temperature (T_pc_) and melting enthalpy (ΔH) of BW oleogel-based margarine increased with the increase in BW content, due to the existence of higher-melting-point BW crystals. For instance, The T_pc_ and ΔH of 4% BW oleogel-based margarine was 46.76 °C and 3.99 J/g. When the BW content increased to 10%, the Tpc and ΔH reached 52.23 °C and 9.52 J/g. As a result, the BW oleogel-based margarine samples could not be melted completely at oral temperature (<37 °C), which reflected the disadvantages of BW oleogel in the aspect of taste and the limitations of application of BW oleogel in the food industry [13]. 

Figure 2B–D shows the change of G’ and G’’ with the temperature sweep. The G’ and G’’ of commercial margarine decreased rapidly with the increase in temperature from 26 °C to 40 °C, and the temperature transition point (G’ = G’’) was around 40 °C, indicating that the internal structure of commercial margarine was gradually destroyed from 30 °C and completely collapsed at 40 °C, with no TAG crystals appearing at this temperature, as shown in Figure 1B (SFC = 0% at 40 °C). Combined with the SFC and DSC results, it can be concluded that the commercial margarine had good mouth-melting properties. 

On the other hand, the temperature transition points of BW oleogel-based margarine with different BW concentrations were all higher than 40 °C, and the higher temperature transition point related to high-melting-point waxes in the oleogel-based margarine. It is important to note that the internal network structure of oleogel-based margarine could not completely collapse under human body temperature, and exhibited the characteristics of a waxy taste. In addition, the change of G’ and G’’ with the temperature sweep had a similar trend with the DSC melting curve, and would not be disturbed by the water phase components. Furthermore, temperature sweep and temperature transition point (G’ = G’’) can be used to evaluate the mouth-melting properties of oleogel-based margarine.

### 2.5. Microstructure and Polymorphism

The bright-field and polarized light microscopy (PLM) microstructure of BW oleogel-based margarine and commercial margarine at 20 °C are shown in Figure 3. As expected, oleogel-based margarine prepared with a rapid cooling process had a dense droplet aggregation structure, which was similar to that of commercial margarine under bright-field light. The density of droplets and crystal network structure increased with the increment of BW content. This behavior was attributed to the formation of a stronger gel network at the action of more BW crystals. However, droplet aggregation appeared in the oleogel-based margarine prepared without a rapid cooling process when the BW content was below 4%, and partial aggregation of crystals occurred for all the oleogel-based margarine at different proportions BW, prepared without a rapid cooling process. On the contrary, smaller crystals and denser network structures were observed in the oleogel-based margarine prepared with a rapid cooling process, indicating that the difference in cooling rate resulted in the difference of BW crystals and structured droplets. The results are consistent with previous studies, which pointed out that the crystallinity of BW increased at supercooling, resulting in a more ordered crystal network structure [25]. Therefore, the partial aggregation of crystals led to the aggregation of droplets in non-quenched samples at low BW content, resulting in structural instability, while high crystal concentration increased the hardness and viscosity of the margarine, which would destroy the ductility of the margarine samples.

The polymorphism of special fat has a great influence on its structural properties and table spread. In general, TAG crystallized to form three different polymorphic forms, that is *α*, *β*, and *β′*. The *β′* form is an ideal crystal form for producing special fat with a good mouthfeel and smooth texture [26]. Figure 4 shows the polymorphism of BW oleogel-based margarine and commercial margarine. As shown in Figure 4, the XRD patterns of BW oleogel-based margarine with different BW contents had the *β′* polymorphic form with peaks of 4.20 Å and 3.70 Å [27]. The crystallinity increases with the increase in BW content due to the formation of more BW crystals. According to our previous study [28], the polymorphic forms of oleogel depended on the composition and structure of wax. The van der Waals force interaction between the hydrocarbon chain and the planar layered lactone group in BW resulted in the formation of vertically orthogonal sub-crystal cell morphology, which is shown as *β*’ form. Furthermore, the processing conditions did not affect the polymorphic formation of oleogel-based margarine, with *β*’ form appearing in the samples prepared with or without a rapid cooling process. The commercial margarine had a signal peak near 42 Å and 14 Å in the SAXD area, indicating the formation of a double-chain length stacking structure, and a signal peak near 4.6 Å in the WAXD area with *β* polymorphic form. From the perspective of polymorphism, oleogel-based margarine with *β*’ form had more advantages than commercial margarine with *β* polymorphic form. However, the long-chain components in BW not only provided an ideal crystal form but also resulted in a high melting point of samples, which made the margarine prepared by BW oleogel unable to have similar macro functional characteristics with commercial margarine.

### 2.6. Texture and Spreadability of Binary Wax Oleogel-Based Margarine

Previous research showed that the BW oleogel-based margarine differs significantly from commercial margarine in physical properties. The crystallization network of BW could not provide suitable mechanical properties and melting characteristics. Therefore, the binary compound oleogel (BW50:ZLW50) was further selected and studied, aiming to achieve a good appearance and properties of margarine.

The hardness of binary wax-based margarine is presented in Figure 5A. The hardness of quenched binary wax-based margarine was significantly lower in comparison to the non-quenched group, indicating that slow crystallization tends to form relatively coarse crystals. In addition, the commercial margarine had a higher hardness than that of 5% binary wax-based margarine. Importantly, the different hardness of binary wax-based margarine between the non-quenched and the quenched sample at the same proportion was smaller than that of the BW oleogel-based margarine. According to our previous research, BW was composed of long-chain wax ester or alkanes, which could form a stronger needle-like crystal network structure, resulting in better plastic characteristics, while the main composition of triglycerides in ZLW led to the formation of a weaker needle-like crystal network, meeting specific formulation criteria in mouthfeel. For the binary wax (BW50:ZLW50), BW crystallized and formed a dense network structure, and then ZLW filled into the network structure of BW. BW and ZLW with similar crystal sizes enhanced the order of the network structure of oleogels, showing both suitable mechanical properties and suitable mouthfeels [17]. Therefore, the crystallization network of oleogel-based margarine could be regulated through the interaction of BW and ZLW, providing softer texture characteristics.

The spreadability of binary wax-based margarine is displayed in Figure 5B. The BW50:ZLW50-based margarine prepared with a rapid cooling process showed good spreadability, while the 5% BW50:ZLW50-based margarine prepared without a rapid cooling process showed a translucent gel form with phase separation. These findings indicated that the rapid cooling process was conducive to the formation of a stable crystalline network of binary wax to structure the aqueous phase. However, the excess wax in the margarine caused sanding during the application, resulting in insufficient sensory properties [23].

### 2.7. Rheological Properties of Binary Wax Oleogel-Based Margarine

Figure 5C–E shows the rheological properties of binary wax-based margarine. The mechanical properties of binary wax-based margarine showed a similar trend to that of BW-based margarine. The LVR of margarine prepared with a rapid cooling process at the same wax content was slightly higher than that of the samples without a rapid cooling process, and the LVR decreased with increasing wax content. The change of G’ and G’’ with frequency scan showed that the G’ of the non-quenched binary wax margarine was higher than that of the quenched samples. The rheological properties of 5% BW50:ZLW50 margarine prepared with a rapid cooling process were similar to that of commercial margarine, which may be relevant that the interaction of BW crystal network and ZLW crystal network in the mixture system of BW50:ZLW50 [17]. Additionally, as shown in Figure 5E, the temperature transition point (G’ = G’’) of 5% BW50:ZLW50-based margarine was lower than 40 °C, indicating its good mouth-melting properties. These results suggest that the macroscopic mechanical properties of margarine could be affected by regulating the crystallization of BW by adding ZLW.

### 2.8. Microstructure of Binary Wax Oleogel-Based Margarine

The bright-field and polarized light microscope images of binary wax-based margarine are shown in Figure 6. Binary wax-based margarine prepared with rapid cooling processes had a dense droplet aggregation structure, which was similar to that of commercial margarine under bright light. However, 5% BW50:ZLW50-based margarine prepared without rapid cooling processes showed bigger droplet size and droplet aggregation patterns with different sizes, indicating that this system was unstable and phase separation occurred. With the increase in BW50:ZLW50 content, the droplet size decreased, and more uniform droplets formed a relatively stable system. In addition, PLM showed that smaller crystals and denser crystal networks were observed in binary wax-based margarine prepared with a rapid cooling process. Therefore, with the increase in the mass fraction of wax under rapid cooling processes, a stronger micro three-dimensional network construction formed to entrap more liquid oil and water, forming more stable margarine. For example, smaller crystals and denser network structures appeared in binary wax-based margarine prepared with a rapid cooling rate, compared with that of non-quenched oleogel-based margarine with the same content of wax (Figure 6). These findings indicated that ZLW could regulate the crystallization of BW to form a uniform network for margarine.

## 3. Materials and Methods

### 3.1. Materials

BW with a melting point (MP) of 48 °C was purchased from Likangweiye Technology Co., Ltd. (Beijing, China). China lacquer wax (ZLW) with an MP of 33.2 °C was purchased from Wankesi Technology Co., Ltd. (Tianjin, China). High-oleic sunflower oil was obtained from the local supermarket. Mono-diglycerol fatty acid ester (DS), soybean lecithin, and whey protein isolate were purchased from Sigma-Aldrich (Shanghai, China). 

The physicochemical properties of waxes were as follows; the high-melting-point wax was BW with 58% wax esters, 26.84% hydrocarbons, 8.75% free fatty acids, and 0.32% free fatty alcohols. The low-melting-point wax was ZLW, which is mainly composed of triglycerides, including more than 80% fatty acids, 2% free fatty acids, and less free fatty alcohols. The above data mainly come from our previous report [17].

### 3.2. Preparation of Margarine

The preparation of oleogel was referred to in our previous research [17]. The linseed oil and BW were mixed at different proportions. The concentrations of BW added to these blends were 4% wt, 6% wt, 8% wt, and 10% wt. The mixed samples were magnetically stirred at 90 °C using a heating plate for 30 min to achieve complete melting. Thereafter, the heated solutions were cooled at 5 °C to produce the stable oleogel.

The preparation of oleogel-based margarine was as follows: Firstly, the oil phase was prepared by adding 0.2%wt DS and 0.1%wt soy lecithin into the completely melted wax oleogel and mixing using a homogenizer (10,000 rpm/min for 1 min). The aqueous phase was prepared by adding 1.6%wt whey protein isolate and 2% wt salt into the water and stirring until the ingredients were dissolved. Then, the oil phase and water phase were mixed at a ratio of 80:20 (wt/wt), and then a homogenizer at 10,000 rpm/min was used to shear and emulsify in an ice water bath (quenched sample) or room temperature (non-quenched sample) for 3 min. After that, the sheared emulsions were stirred slowly for 5 min. Then, the samples were stored in a 4 °C incubator before any analysis.

### 3.3. Hardness

The hardness of margarine was determined using a TA-XT2i Texture Analyzer (Stable Micro Systems, Surrey, England). The probe P5/N was used with the Pre-test speed of 1.00 mm/s, test speed of 2.00 mm/s, post-test speed of 2.00 mm/s, trigger force of 5.0 g, and puncture depth of 12.00 mm. Each sample was measured 6 times, and the average force was reported as firmness.

### 3.4. Rheology

Rheological measurements of oleogel-based margarine were performed by use of a DHR-3 rheometer (Waters Instruments, Worcester County, MD, USA) equipped with a 40 mm diameter parallel plate with a gap of 1000 μm. The Peltier system for temperature control was used for temperature testing. To determine the properties of the linear viscoelastic region (LVR), amplitude scanning was performed at 1 Hz and a strained amplitude range from 0.001 to 100%. The frequency sweep was applied from 0.1 to 10 Hz, and the strain was controlled at 0.01% [29]. The storage modulus (G’) and loss modulus (loss modulus, G’’) of the sample at different temperatures (at a rate of 5 °C/min from 0 °C to 80 °C) were measured using an oscillating stress of 1 Pa and a frequency of 1 Hz.

### 3.5. Morphology

The microstructure of the margarine was photographed using optical and polarized light microscopy (DM2700P, Leica, Germany) equipped with a Leica DFC450 video camera (Leica, Germany). The melted samples were placed on a preheated glass slide and the coverslip was placed on the surface of the samples. After that, samples were stored in the temperature-controlled incubator for 24 h before observation under the microscope.

### 3.6. Solid Fat Content

The solid fat content (SFC) of margarine was analyzed using PC-120 pulsed nuclear magnetic resonance (pNMR) (Bruker, Karlsruhe, Germany). Approximately 3 mL of molten oleogel samples were filled into pNMR tubes and heated in a water bath at 80 °C for 30 min to remove the crystal memory. Then, the samples were kept at 0 °C for 1 h to crystallize completely. After that, it was determined at different temperatures from 0 °C to 60 °C by a 5 °C increase. Each sample was determined at 30 min intervals.

### 3.7. Polymorphism

D2 Phaser X-ray diffraction (XRD) (Bruker, Karlsruhe, Germany) was used to determine the long and short-spacing diffraction patterns of the sample with Ni filters and Cu-K radiation resistance with the divergence slit, scatter slit, and receiving slit of 1.0, 1.0, and 0.3 mm, respectively (k = 1.54056 Å, current 40 mA, voltage 40 kV). The sample was scanned from the angle of 1° to 30° at a rate of 0.05°/min at ambient temperature. Data were analyzed using SA Data Inc., Livermore, CA, USA.

### 3.8. Thermal Properties

The thermal properties of margarine were measured using the differential scanning calorimeter (DSC 8500, Perkin Elmer, Waltham, MA, USA). Samples were prepared by placing 3–8 mg of oleogel into the crucible. The empty crucible was used as a control. Firstly, the sample was heated at 80 °C for 10 min until it was completely melted to remove the crystal memory, and then the temperature was cooled to 0 °C at 5 °C/min to analyze the crystallization behavior. The sample was kept at 0 °C for 10 min to reach equilibrium and then heated to 80 °C at 5 °C/min to analyze its melting behavior.

### 3.9. Statistical Analysis

The experimental data were analyzed using one-way ANOVA in the data analysis software SPSS 19.0. The analysis of differences between multiple sets of samples was obtained by Duncan model testing. Statistical analysis was performed using Origin 9.4 software. 

## 4. Conclusions

In this study, oleogel-based margarine was prepared, and its physical properties and microstructure were compared with commercial margarine. The hardness of BW oleogel-based margarine depends on the wax concentrations. The BW oleogel-based margarine prepared with a rapid cooling process had much higher hardness than that of margarine prepared without a rapid cooling process. The oleogel-based margarine with different contents of BW had a similar LVR, which was shorter than that of commercial margarine. The DSC study showed that the BW oleogel-based margarine had a higher melting peak at 40 °C than that of 33.45 °C of commercial margarine, leading to the higher temperature transition point of BW oleogel-based margarine, resulting in its waxy taste. XRD patterns of oleogel-based margarin with different content BW were quite similar and structurally to the β′ form. According to PLM images, the denser droplets and crystal network structure was observed with the increment of BW content. However, the higher content of BW led to the margarine’s bad mouth-melting properties. Furthermore, the unique, improved physical properties of oleogel-based margarine were obtained with binary mixtures of ZLW and BW. The crystallization network of oleogel-based margarine was regulated due to the interaction of the BW and ZLW crystal network with the enhancement of the order of network structure in oleogels. In addition, the rapid cooling process improved the spreadability of oleogel-based margarine. As a result, the 5% BW50:ZLW50 oleogel-based margarine prepared using a rapid cooling process had a good appearance and similar mechanical properties to that of commercial margarine, especially with good mouth-melting propeties (37 °C), which had the potential to replace the commercial margarine.

## Figures and Tables

**Figure 1 molecules-27-08952-f001:**
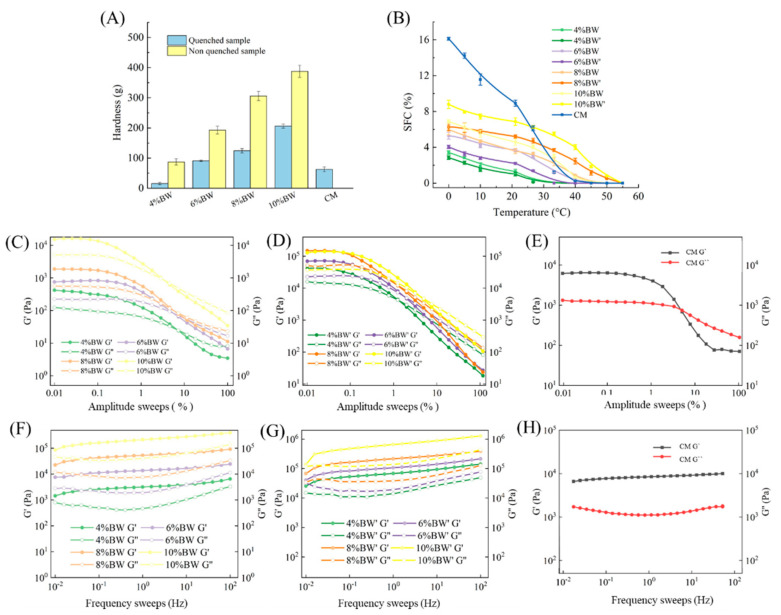
Hardness (**A**), SFC, DSC (**B**), amplitude sweeps (**C**–**E**), and frequency sweeps (**F**–**H**) of BW oleogel-margarine and commercial margarine.

**Figure 2 molecules-27-08952-f002:**
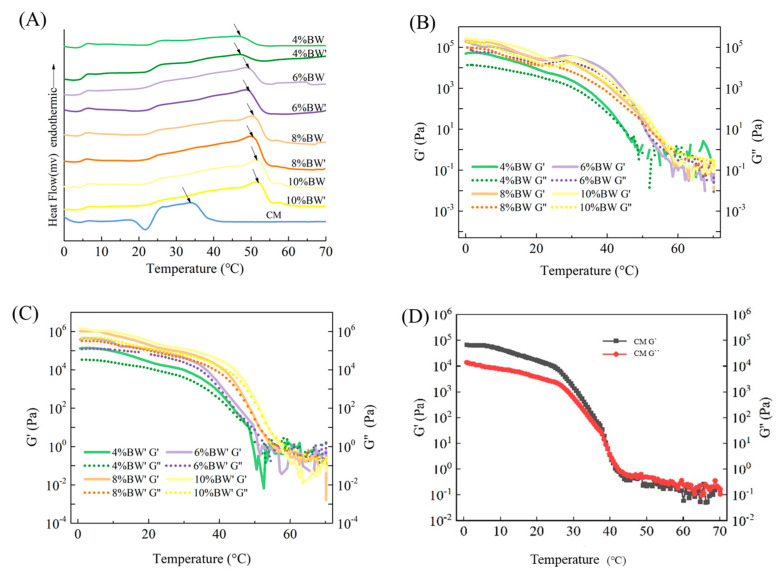
DSC melting curves (**A**), temperature ramps of BW oleogel-based margarine (**B**,**C**), and commercial margarine (CM) (**D**), respectively.

**Figure 3 molecules-27-08952-f003:**
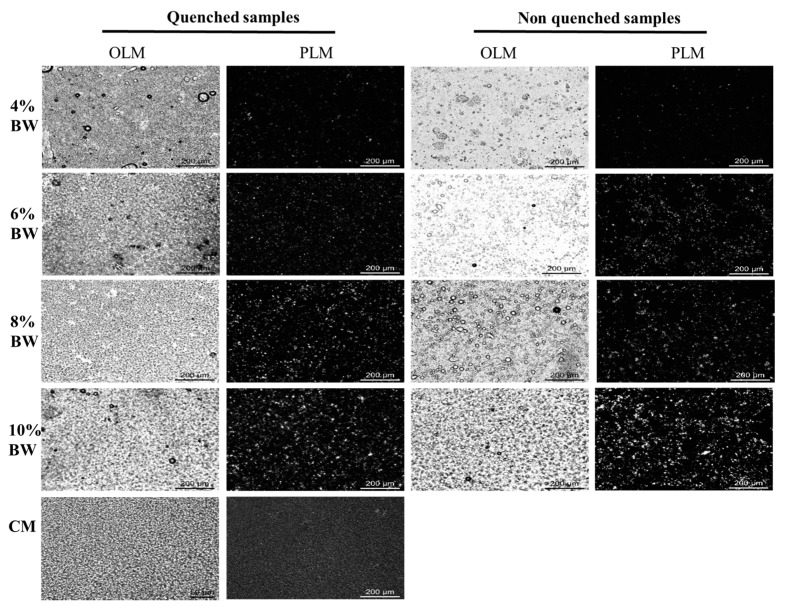
Bright-field and polarized light images of BW oleogel-based margarine and commercial margarine, respectively.

**Figure 4 molecules-27-08952-f004:**
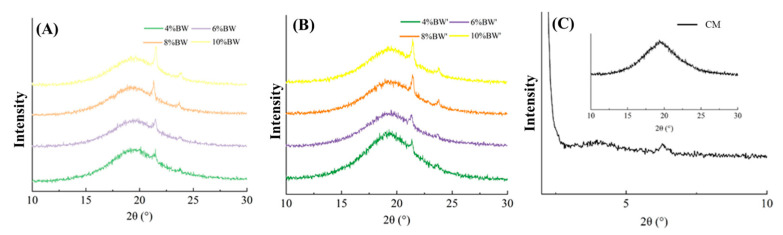
XRD patterns of BW oleogel-based margarine (**A**,**B**) and commercial margarine (C), respectively.

**Figure 5 molecules-27-08952-f005:**
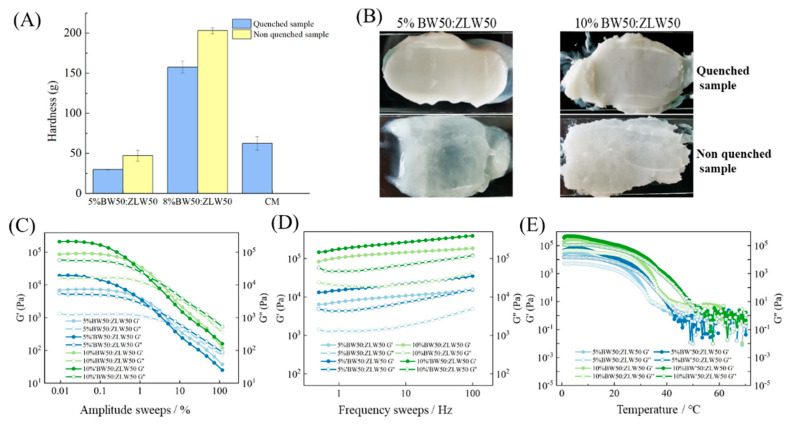
Hardness (**A**), smear appearance (**B**), amplitude sweeps (**C**), frequency sweeps (**D**), and temperature ramps (**E**) of binary wax-based margarine, respectively.

**Figure 6 molecules-27-08952-f006:**
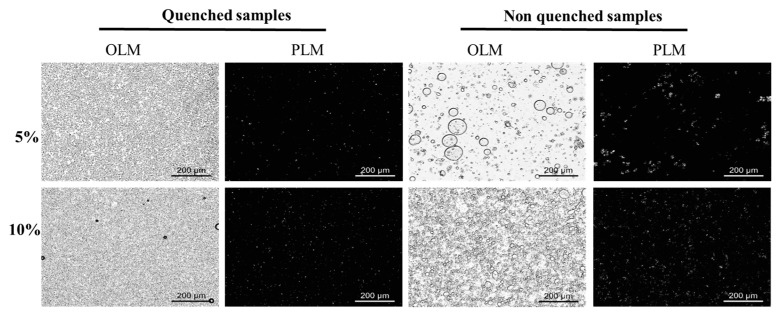
Bright-field and polarized light images of BW50:ZLW50 margarine, respectively.

## Data Availability

All data are available in the manuscript and Appendix A.

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
