# Peer review of "Crystallization and Structural Properties of Oleogel-Based Margarine"

_molecules, 2022, doi:10.3390/molecules27248952_

Round 1
Reviewer 1 Report
Manuscript Number: Molecules 2051716
The manuscript written by Chai et al. titled “Crystallization and structural properties of oleogel-based margarine” provided a detailed study of the feasibility of using oleogel in compassion to commercial margarine. I want to suggest the following comments and clarify a few concerns which will further improve the article and make it suitable for molecules.
Comments
1. Please check on the abbreviations, especially for the China Lacquer Wax. Someplace it CLW and other times it's ZLW.
2. Please include the cooling ramps for the formulations from DSC analysis, and include the crystallization energy. Sometime from the crystallization peak behavior one can tell a lot about the homogeneity of the crystals and relative % crystallization.
3. Is there a specific reason for choosing 5 °C for the cooling of the oleogels? It is very important as different cooling rates might vary the crystallization and the final properties of the oleogels.
4. I don’t think doing XRD was necessary, it might be excluded or put in the supplementary.
5. Please check other grammar mistakes like commas, periods, etc.
Reviewer 2 Report
The manuscript by Liu et al. suggests that 5% BW50:ZLW50 has similar properties to commercial margarine. The authors have made several experimental comparisons with margarine (e.g., rheology) and between quenched and non-quenched samples.
The manuscript needs the following revisions:
Technical:
- The authors need to label the figure captions in the text properly. A few examples include: a) Figure 10 (line281) and Figure 7 (lines 217,218) are non existent.
-There is not optical microscopy image showing the crystals (see for instance, previous ref. 17)
- The authors mentioned a flow sweep test in the methods, but the results are not shown. Please verify this.
- Included dashed lines for G" in Figure 2.
- The sentence "Figure 2 shows the G'/G'' changes of samples with the frequency sweep" is unclear. Figure 2 shows temperature ramps at a constant frequency (1 Hz)
- Include the volume used for the preparation of oleogels and margarine (section 3.2)
- In Figure 2A, it is unclear where the 33.45 deg C peak for margarine comes from. Please include arrows to highlight those values. What is the peak ca. 20 deg C?
Editorial:
- Please check the text. The authors used olegel, oleogel, and olegels, many times throughout the text.
- Add a capital letter after a complete stop sign.
- "studys"is not a word
- Add a space between symbols (line 216 for Beta) and a space between temperature signs. For example, 20 °C instead of 20°C (line 164 to mention one instance)
- Please make fig. 1 larger. It is hard to read
- The authors mentioned: "indicating that the difference in cooling rate resulted in the difference of BW crystals and structured droplets" Was there a DSC performed between quenched and unquenched samples? Fig. 2A only correlates with concentration. Please clarify
- Define acronyms in the abstract (BW, CLW, ZLW)
- Can the authors expand their explanation on the eutectic properties of BW50:ZLW50? For instance, refer to lines 273 and 274. This discussion can be expanded with the data presented. In particular, if it is a major conclusion of the study.
Round 2
Reviewer 1 Report
Thank you for making the corrections. Please make sure that there is a gap between the number and the unit, like 40 °C.
Reviewer 2 Report
Thank you for addressing all the comments. Prior publication make sure to check the word "oleogel" very carefully. There are two instances where it is mentioned oleogle . Please make sure to edit it before final publication
lines 335 and 412